# Comprehensive Profiling of Mammalian Tribbles Interactomes Implicates TRIB3 in Gene Repression

**DOI:** 10.3390/cancers13246318

**Published:** 2021-12-16

**Authors:** Miguel Hernández-Quiles, Rosalie Baak, Anouska Borgman, Suzanne den Haan, Paula Sobrevals Alcaraz, Robert van Es, Endre Kiss-Toth, Harmjan Vos, Eric Kalkhoven

**Affiliations:** 1Center for Molecular Medicine, University Medical Center Utrecht, Utrecht University, 3584 CG Utrecht, The Netherlands; M.HernandezQuiles-2@umcutrecht.nl (M.H.-Q.); r.e.baak@students.uu.nl (R.B.); A.Borgman-3@umcutrecht.nl (A.B.); s.l.haan@students.uu.nl (S.d.H.); 2Oncode Institute and Molecular Cancer Research, Center for Molecular Medicine, University Medical Center Utrecht, Utrecht University, 3584 CG Utrecht, The Netherlands; P.SobrevalsAlcaraz-2@umcutrecht.nl (P.S.A.); R.M.vanEs-4@umcutrecht.nl (R.v.E.); H.R.Vos-3@umcutrecht.nl (H.V.); 3Department of Infection, Immunity and Cardiovascular Disease, Medical School, University of Sheffield, Sheffield S10 2TN, UK; e.kiss-toth@sheffield.ac.uk

**Keywords:** tribbles, proteomics, interactome, breast cancer

## Abstract

**Simple Summary:**

Tribbles proteins play various roles in cancer initiation and progression. However, still little is known about their molecular actions. Here we developed a mass spectrometry-based approach to study the Tribbles interactomes, allowing us to discover new interactors and functions that might help to understand their behavior better. Our proteomics data highlight the ability of TRIB3 to interact with transcription regulatory proteins and point to a new role in gene repression. Systematic analyses like these will help to evaluate the potential of Tribbles proteins as biomarkers for disease diagnosis and prognosis.

**Abstract:**

The three human Tribbles (TRIB) pseudokinases have been implicated in a plethora of signaling and metabolic processes linked to cancer initiation and progression and can potentially be used as biomarkers of disease and prognosis. While their modes of action reported so far center around protein–protein interactions, the comprehensive profiling of TRIB interactomes has not been reported yet. Here, we have developed a robust mass spectrometry (MS)-based proteomics approach to characterize Tribbles’ interactomes and report a comprehensive assessment and comparison of the TRIB1, -2 and -3 interactomes, as well as domain-specific interactions for TRIB3. Interestingly, TRIB3, which is predominantly localized in the nucleus, interacts with multiple transcriptional regulators, including proteins involved in gene repression. Indeed, we found that TRIB3 repressed gene transcription when tethered to DNA in breast cancer cells. Taken together, our comprehensive proteomic assessment reveals previously unknown interacting partners and functions of Tribbles proteins that expand our understanding of this family of proteins. In addition, our findings show that MS-based proteomics provides a powerful tool to unravel novel pseudokinase biology.

## 1. Introduction

Kinases regulate a plethora of cellular processes and changes in their enzymatic activity are intimately linked to human diseases, hence the large research field studying the basic biology of kinases and their potential as therapeutic targets [1,2]. In addition to 518 kinases, the human genome also encodes ~60 pseudokinases, proteins that resemble serine/threonine and tyrosine protein kinases but lack several amino acids critical for enzymatic activity [3,4]. The human pseudokinase family includes the three members of the Tribbles (TRIB) family—TRIB1, TRIB2 and TRIB3—that share a high degree of homology as well as similar domain compositions [5,6]. They can be divided into three major domains: an N-terminal domain, associated with protein stability and subcellular localization [7,8]; a well-conserved, centrally located pseudokinase domain; and a C-terminal domain, wherein the binding motifs of MAPK and COP1 are found [9,10]. A fourth, more distally related protein, called STK40, shares important similarities in terms of function and structure [11,12].

Tribbles proteins have been implicated in multiple critical signaling and metabolic processes and alterations in their expression and/or activity is linked to various human diseases [13]. While lacking intrinsic enzymatic activity, Tribbles proteins exert their biological roles predominantly via binding to other proteins, including kinases, phosphatases, transcription factors and components of the ubiquitin-proteosome system [14,15,16]. This diverse range of interactors explains, at least in part, the difficulties to associate a TRIB family member with a single specific cellular pathway or role. In addition, it should be noted that different and even contradictory observations have been made regarding the subcellular localization of Tribbles proteins, suggesting their localization and thereby function depends on cellular context and conditions [17,18].

In recent years many studies have pointed to Tribbles proteins as important modulators of cancer initiation and progression [19,20,21,22]. Therefore, Tribbles proteins hold potential as biomarkers of disease diagnosis and prognosis as well as pharmaceutical targets for a number of cancers [23]. For example, TRIB1 upregulation is significantly associated with metastasis and poor prognosis in prostate cancer [24], it has been shown that TRIB1 mediates radioresistance in glioma cells by an HDAC1-dependent pathway [25] and high levels of TRIB1 are associated with poor breast cancer survival through the regulation of PI3K-NFκB pathway [26]. TRIB2 has been shown to contribute to tumorigenesis in lung cancer through the downregulation of C/EBPα [27] and TRIB2 direct interaction with AKT has been shown to be an important mechanism that contributes to resistance to anti-cancer drug therapy [28]. Finally, TRIB3 has been shown to support breast and colorectal cancer stemness through the interaction with AKT and beta-catenin respectively [29,30]. These examples illustrate that the different Tribbles family members can all play a regulatory role in cancer initiation and progression, but their contribution may be tumor type specific. Furthermore, these examples also add to anecdotal evidence that critical interacting proteins may differ between Tribbles family members and to previous reports that the affinities of distinct TRIB proteins to the same binding partner may differ [31]. We hypothesize, therefore, that particular Tribbles functions are dictated by its interactome—the specific set of proteins with which a tribble family member is interacting within a given biological setting—and that improving our understanding of how these interactions take place will help to define the roles of Tribbles proteins in each context. To date, comprehensive Tribbles interactomes have not been reported. 

Mass spectrometry (MS)-based proteomics approaches have been widely used in recent decades to study and identify protein–protein interactions (PPIs) [32]. Affinity-purification mass spectrometry (AP–MS) is used for the purification of a protein (endogenous or tagged) and its interacting partners from a cell lysate [32]. This technique relies on the affinity of an antibody (or nanobody) for a protein and is followed by MS analysis [32]. We have previously used this approach successfully to identify the interactomes of various intracellular proteins [33,34,35]. 

In this study we have developed a robust AP-MS approach to characterize the TRIB1, -2 and -3 interactomes. In addition, we have investigated the contribution of the different domains of TRIB3 to its interactome. Finally, we have generated an inducible system to evaluate the similarities and differences between TRIB1 and -3 interactomes in breast cancer cells as a first proof-of-principle study showing that comprehensive profiling of interactomes can improve our understanding of Tribbles’ role in cancer onset and progression.

## 2. Materials and Methods

### 2.1. Materials

Primary antibodies anti-turboGFP (Origene, #TA150041, Rockville, MD, USA), anti-Gal4DBD (Santa Cruz, sc-510, Santa Cruz, CA, USA) and anti-tubulin (Sigma Aldrich, T9026, St. Louis, MO, USA) were used. Secondary antibodies anti-mouse-HRP (Dako, P0260, Glostrup, Denmark) and anti-rabbit-HRP (Thermofisher, #31460, Waltham, MA, USA) were used. GFP-Trap and tGFP-Trap Agarose beads (Chromotek, Planegg, Germany) were used for immunoprecipitation. Doxycycline (Sigma Aldrich, D9891).

### 2.2. Cell Culture

Human HEK293T embryonic kidney cell line (ATCC CRL-3216, Manassas, VA, USA) and human MCF7 breast cancer cells (ATCC HTB-22) were maintained in DMEM 4.5 g/L d-glucose supplemented with 10% fetal bovine serum and 1% penicillin and streptomycin. Cells were incubated in 5% CO_2_ incubator at 37 °C and 95% humidity. To generate cells stably overexpressing TRIB1 or -3, MCF7 cells were transduced with third-generation lentiviral constructs using supernatants from HEK293T cells transfected with lentiviral packaging plasmids. HEK293T cells were transfected using X-treme gene 9 DNA transfection reagents (Roche), according to the manufacturer’s protocol.

### 2.3. Plasmids

TRIB3 and TRIB1 expression plasmids were kindly provided by Dr. Endre Kiss-Toth. Pcw57.1 lentiviral construct was provided by Dr. S.W.C. van Mil (UMC Utrecht, Utrecht, the Netherlands). Deletions of the N- and C-terminal regions of TRIB3 were performed using a Quickchange mutagenesis kit (Stratagene, San Diego, CA, USA). Successful mutagenesis was verified by Sanger sequence analysis. The reporter construct 5xGAL4-TK-Luc-pGL3 has been described previously [36]. The pCDNA-Gal4DBD-TRIB3 was generated by cloning TRIB3 BamH1/Xba1 fragment from TRIB3 expression plasmid into the respective sites of pCDNA-Gal4DBD as described before [37]. pCDNA-Gal4DBD-TRIB3-ΔN-terminal (amino acids 69 to 358) and pCDNA-Gal4DBD-TRIB-ΔC-terminal (amino acids 1 to 316) were generated using Quickchange mutagenesis kit (Stratagene).

### 2.4. Luciferase Reporter Assays

HEK-293T and MCF7 cells were transfected using Xtreme gene 9 DNA transfection reagent (Roche) in 24-well plate format; 100 ng pCDNA3.1-Gal4DBD-TRIB3WT and mutants, 1 μg of pGL3 reporter and 2 ng of TK-Renilla luciferase were used for the experiments. After 48 h cells were lysed and firefly and Renilla luciferase were measured with a Dual-Luciferase Reporter Assay System (Promega, Madison, WI, USA) in a TriStar2 LB942 Multimode Reader (Berthold Technologies, Bad Wildbad, Germany). The results are expressed in relative luciferase units; the results are an average of three independent experiments. Student’s *t*-tests were used. Statistical significance was defined as *p* < 0.05.

### 2.5. Western Blot Analysis

Western blotting was performed as described before [37] In short, cells were grown in 6-well format or 10-cm dishes. After induction with 2 μg/mL doxycycline for 24 h, cells were washed with ice-cold PBS, twice, and scraped in ice-cold lysis buffer (150 mM NaCl, 1% NP40, 0.5% sodium DOC, 0.1% SDS, 25 mM Tris pH 7.4) supplemented with protease inhibitors. After incubation on ice for 20′, samples were centrifuged at maximum speed for 10′ at 4 °C and supernatants were collected. Protein concentrations were measured, samples were supplemented with Laemmli Sample Buffer (LSB) and incubated at 95 °C for 5′ before use. Samples were separated by SDS-PAGE and then transferred to PVDF membrane. Blocking was performed in 5% milk in TBS-T for 45′. Incubation with primary antibody was done overnight at 4 °C and secondary for 1 h at room temperature. Membranes were treated with ECL Western blot solution and protein expression was detected using LAS4000 Image Quant.

### 2.6. Immunoprecipitation

HEK293T cells were seeded in 15 cm dishes and transfected when the cells were approximately at 80% confluency. Forty-eight hours after transfection cells were washed in ice-cold PBS twice and then scrapped in 2 mL of lysis buffer (50 mM Tris 8.0 pH, 1 mM EDTA pH 8.0, 0.1% NP40, 250 mM NaCl, 10% Glycerol). Samples were incubated on ice for 20′ and then spun down at maximum speed for 10′ at 4 °C, supernatant was collected. GFP-Trap beads (Chromotek) were equilibrated according to the manufacturer’s protocol and incubated with the supernatant from the previous step for 2 h at 4 °C. Then beads were collected by spinning down the samples at 2500 g for 5′ at 4 °C. Beads were washed two times with lysis buffer and one final time in PBS before being transferred to a low-binding Eppendorf tube. Finally, the beads were spun down and dried using a Pasteur pipette. MCF7 cells were induced with 2 μg/mL doxycycline for 24 h before lysis and incubation with turboGFP-Trap beads (Chromotek) as described above.

### 2.7. Confocal Microscopy

Cells were grown in μ-Slide 8-well glass-bottom chambers (Ibidi) and treated with 2 μg/mL doxycycline for 24 h. Cells were incubated with DAPI (Vectashield) for 10 min to stain nuclei. Images were obtained using LSM880 Zeiss Microscope.

### 2.8. Mass Spectrometry

The precipitated proteins were denatured and alkylated in 50 µL 8 M Urea, 1 M ammonium bicarbonate containing 10 mM tris (2-carboxyethyl) phosphine hydrochloride and 40 mM 2-chloro-acetamide. After 4-fold further dilution with 1M ammonium bicarbonate and digestion with trypsin (250 ng/200 µL), peptides were separated from the sepharose beads and desalted with homemade C-18 stage tips (3 M, St Paul, MN, USA). Peptides were eluted with 80% ACN and, after evaporation of the solvent in the speedvac, redissolved in buffer A (0.1% formic acid). After separation on a 30-cm pico-tip column (75 µm ID, New Objective) in-house packed with C-18 material (1.9 µm aquapur gold, dr. Maisch) using a 140-min gradient (7% to 80% ACN, 0.1% FA), delivered by an easy-nLC 1000 (Thermo), peptides were electro-sprayed directly into an Orbitrap Fusion Tribrid Mass Spectrometer (Thermo Scientific). The MS was run in DDA mode with a cycle time of 1 s, in which the full scan (400–1500 mass range) was performed at a resolution of 240,000. Ions reaching an intensity threshold of 10,000 were isolated by the quadrupole and fragmented with an HCD collision energy of 30%. 

The obtained raw data was analyzed with MaxQuant [version 1.6.3.4], using the Uniprot fasta file (UP000005640) of Homo sapiens (taxonomy ID: 9606), extracted at 21/01/2021. Minimum and maximum peptide lengths of 7 and 25 amino acids respectively, with Oxidation on Methionine and Acetylation on Protein N-term as variable modifications and Carbamidomethyl on Cysteine as a fixed modification. Peptide and protein false discovery rates were set to 1%. 

To determine proteins of interest, we performed a differential enrichment analysis on the generated Maxquant output. First, we generated unique names for the genes associated to multiple proteins to be able to match them. Second, we filtered for proteins that were identified in at least three out of four of the replicates of one condition. Then, we background corrected and normalized the data by variance stabilizing transformation; shifting and scaling the proteins intensities by sample group. We used a left-shifted Gaussian distribution to impute for missingness, since our data presented a pattern of missingness not at random (MNAR). Finally, we performed a differential enrichment analysis to identify those proteins that were over-enriched and selected those with at least a 2.5-fold change and adjusted *p*-value ≤ 0.05. The adjusted *p*-value was calculated using the Benjamin–Hochberg procedure. The program used for the analyses was R [version 4.0.4] through R-Studio [version 1.5.64]. The mass spectrometry proteomics data have been deposited to the ProteomeXchange Consortium via the PRIDE partner repository (Available online: http://www.ebi.ac.uk/pride (accessed on 15 December 2021). Dataset identifiers will be provided during review.

## 3. Results

### 3.1. Analysis of TRIB1, TRIB2 and TRIB3 Interactomes in HEK293T Cells Using AP-MS

To better understand the distinct role of each Tribbles family member, as well as their redundancies, we developed a method to robustly identify Tribbles interactomes. We transiently overexpressed TRIB1, TRIB2 and TRIB3 as GFP-fusion proteins in HEK293T cells and performed AP-MS experiments. We used nanobodies against GFP to purify Tribbles proteins and their interacting partners, which allowed us to reduce background binding and minimize the amount of peptides released during on-bead digestion [38]. After purification we used liquid chromatography-tandem mass spectrometry (LC–MS) to characterize Tribbles interactomes (Figure 1A).

Among the different interactors that were found we could detect the known binding partners of TRIB1, TRIB2 and TRIB3 as well as interactors that have not yet been reported. The results are described in Table 1, Table 2 and Table 3. As reported before [9,39], all Tribbles family members were able to physically interact with the E3 ubiquitin degradation complex formed by the ubiquitin E3 ligase COP1 and the adaptors proteins DET1 and DDB1. Confirming the specificity of our methodology, we mutated the COP1 binding motif (Figure 1A) and thus specifically depleted the interactome from COP1 and DET1 (Figure 1B,D). TRIB1 and TRIB2 were found to interact with TRIB1 and the Tribbles-related pseudokinase STK40, suggesting that Tribbles proteins can form homo- and heterodimers in mammalian cells, as has been shown for the Drosophila Tribbles homologue Trbl [40] and for the mammalian versions in protein complementation assays (PCA; unpublished observations). Other previously described interaction partners that were detected included activating transcription factor 4 (ATF4) [41]. While not previously described as binding partners, the mTOR regulatory subunits RICTOR and RAPTOR were also detected in the TRIB2 interactome (Figure 1C). Previous studies have shown, however, that TRIB2 regulates mTOR signaling [42,43]. Interactions that have not been described earlier included the interaction between TRIB3 and the mitochondrial transporter TIM-TOM complex, a complex found in the mitochondrial membrane that transports proteins into the inner membrane of the mitochondria [44]. In addition, whilst the interaction between TRIB proteins and the E3 ubiquitin ligase COP1 is well-established [9,45], TRIB1 and TRIB3 showed interaction with another family of E3 ubiquitin ligases, that includes STIP1 and STUB1. Among the most enriched interactors of TRIB2 were SKT and BDH2, two interaction partners, newly identified here, that are involved in the development of different tissues and an enzyme involved in metabolism, respectively [46,47]. In summary, the identification of known interactors of Tribbles validates our experimental approach and provides support for the newly discovered binding partners.

### 3.2. Contribution of the Different Domains to the TRIB3 Interactome

Tribbles proteins contain three distinct domains, with the central pseudokinase domain being the most conserved; the amino acid sequence of human TRIB1/2 and 3 shows a 55% similarity in the pseudokinase domain, but, for example, the C-terminal domains of TRIB3 and TRIB1 are only 9% similar (Appendix A). This low similarity in the N- and C-terminal domains suggest that each holds unique functions and may help to develop TRIB-targeting drugs with low cross-reactivity, but these domains have not been studied intensively.

The N- and C-terminal domains of TRIB3 are unstructured domains for which a 3D conformation cannot be predicted based on the amino acid sequence, as shown in Figure 2A [7]. In order to understand the contribution of the different domains of TRIB3 (N- and C-terminus and pseudokinase domain) to its interactome and to gain an insight into how these interactions take place, we generated TRIB3 mutants lacking the N-terminal domain (amino acids 1–69, TRIB3-ΔN-terminal) and the C-terminal domain (amino acids 316–358, TRIB3-C-terminal) and performed an AP–MS experiment, as described above (Figure 2B). A summary of the top 25 interactions of the C-terminal and N-terminal domains is shown in Table 4. In agreement with previous studies [45], the interaction with COP1 and DET1 requires the presence of the C-terminal domain, as binding is lost with the TRIB3-ΔC mutant (Figure 2C). Moreover, novel proteins related to the ubiquitin degradation system were also found binding to the C-terminal domain of TRIB3, such as UBR2, an E3 ubiquitin-protein ligase that controls cell growth via mTOR signaling [48]. These findings confirm the concept that the C-terminal domain is required for TRIB3 to act as a degradation platform. In addition to the E3 ubiquitin ligases we also identified USP16, a deubiquitinating enzyme that plays an important role in mitosis [49], suggesting a role for TRIB3 in this process as well. In addition, STK40 and DOCK11 were also found to interact through the C-terminal domain of TRIB3. This reinforces the notion that Tribbles proteins and the Tribbles-like protein STK40 can form the homo/hetero dimers mentioned above. DOCK11 is a guanine nucleotide-exchange factor that activates CDC42 and RAC1 [50]; the biological relevance of this interaction remains to be established.

The N-terminal domain of TRIB3 contains a nuclear localization signal sequence that directs TRIB3 towards the nucleus and a PEST domain that might affect TRIB3 stability. The interactors that were lost when deleting the N-terminus included several transcription factors and regulators, which are mainly found in the nucleus, such as ZBTB1, p300, and SPEN (Figure 2C). The interaction between TRIB3 and ZBTB1 was confirmed by co-immunoprecipitation assays (Figure 2D and Appendix A). In addition, we found all the subunits of the WRAD complex (WDR5, DPY30, ASH2L and RbBP5) binding to the N-terminus of TRIB3. The WRAD complex is crucial for SET1 histone methyl transferases to catalyze histone 3 lysine 4 methylation [51]. The interactions between TRIB3 and WRAD complex components were also confirmed by co-immunoprecipitation experiments (to be published elsewhere).

Interestingly we also found the interaction with serine/threonine protein kinase D1 (PRKD1), and the MAPK interacting serine/threonine kinase 1 and 2 (MKNK1 and MKNK2) suggesting that the regulation of mitogen-activated protein kinase (MAPK) signaling does not happen only through the C-terminal [10] and pseudokinase domains [31] as reported earlier but also through the N-terminal domain. It is also worth mentioning the interactions with TP53 and COPS8. TP53, the so-called “guardian of the genome”, is the most common tumor suppressor that is found mutated across all cancer types [52] and it regulates the cell cycle, as well as the apoptosis of damaged cells. COPS8 is a component of the COP9 signalosome that is involved in the phosphorylation of p53 [53].

Taken together, these data indicate that TRIB3 is a putative important transcriptional regulator and this role is carried out mainly through the N-terminal domain. In contrast, the C-terminal domain of TRIB3 is required for the interaction with components of the ubiquitin system and for the formation of homo/hetero dimers.

### 3.3. TRIB3 Function as a Transcriptional Repressor

While our data revealed an interaction between the N-terminus of TRIB3 and for example SET1 histone methyl transferase complexes and p300, which are associated with transcriptional activation [51,54], the same TRIB3 domain also interacted with various proteins that are involved in transcriptional repression, such as ZBTB1 and SPEN [55,56]. In addition, TRIM28 (also known as KAP-1 or TIF1β) and SETDB1, proteins that may form a repressor complex with ZBTB1 [57,58] (were also detected (adjusted *p*-value ≤ 0.05. Data not shown).

To assess the effect of TRIB3 on transcription, we fused TRIB3 to the DNA binding domain of Gal4 (Gal4DBD) and tested the transcriptional activity of the fusion protein on a reporter plasmid with high basal activity(5xGal4-TK-Luc) [36]. As shown in Figure 2E, TRIB3 had a repressive effect when compared to the Gal4DBD alone, similar to for example GalDBD fusions of ZBTB1 and TRIM28 [59,60]. This TRIB3-mediated repression was mostly lost when the N-terminus was deleted, but the TRIB3 ΔC-terminal mutant retained repressor activity (Figure 2E). These results are in line with the mass spectrometry data, described above, in which we found that a high number of transcriptional repressors are able to bind through the N-terminal of TRIB3. Gal4DBD fusion proteins were expressed at similar levels, excluding the possibility that differences in activity were due to expression differences (Figure 2F and Appendix A). We conclude, therefore, that the N-terminus of TRIB3 harbours repressive activity when tethered to the DNA, which may be due to repressor proteins, such as ZBTB1 (and its associated proteins TRIM28 and SETDB1) and SPEN binding specifically to this region of the protein. Furthermore, while the N-terminus of Tribbles has the ability to recruit both transcriptional activators (e.g., MLL complex) and repressors (e.g., ZBTB1 and SPEN), the balance appears to be in favor of transcriptional repression, at least in this experimental setting.

### 3.4. Comparison of TRIB1 and TRIB3 Interactomes in MCF7 Cells

Having developed a robust AP–MS workflow to identify the interaction partners of Tribbles proteins, we wished to address the different roles of TRIB1 and TRIB3 in breast cancer, where high levels of both proteins have been reported to be associated with poor prognosis and lower survival rates [26,61,62]. For this we generated inducible TRIB1-tGFP and TRIB3-tGFP stable cell lines in the breast cancer cell line MCF7, a model for luminal A breast cancer, allowing immediate short-term overexpression of TRIB1 and -3, as well as control over the amount of protein being overexpressed. A summary of the constructs used and the workflow followed to identify interacting proteins is depicted in Figure 3A. Expression of TRIB1-tGFP, TRIB3-tGFP and -tGFP was observed by Western blot after 24 h treatment with doxycycline and no expression was observed in untreated cells (Figure 3B and Appendix A). TRIB3 was predominantly—but not exclusively—localized in the nucleus as determined by confocal microscopy (Figure 3C). In contrast to the predominant nuclear localization of TRIB1 in HEK293T cells ([18] data not shown) TRIB1 was mainly localized in the cytoplasm in MCF7 cells (Figure 3C), supporting the view that subcellular localization of Tribbles depends on cellular context and conditions [17].

Importantly, long-term overexpression of TRIB3 was recently linked to increased proliferation in MCF7 cells [62], which potentially confounds interactome profiles, but no significant effects on cell proliferation were observed within the 24-h timeframe of our experiments (data not shown). A summary of top 20 interactors for TRIB1 and TRIB3 (Figure 3D) is listed in Table 5. Interactors common to both TRIB1 and TRIB3 included COP1 and CDKN1A. The first one is a common interactor of all Tribbles family members that has been widely studied and was also detected in the current study in HEK293T cells (Figure 1), and the second one is a cyclin-dependent kinase inhibitor that is tightly controlled by TP53 [63]. CDKN1A mediates G1 cell cycle arrest in response to a variety of external stimulus. Given the fact that Tribbles were originally described as a cell cycle regulators in Drosophila [64], this might be another mechanism by which these proteins regulate proliferation. In addition, both TRIB1 and TRIB3 also interacted with fatty acid synthetase (FASN), an enzyme that catalyzes the synthesis of palmitate from acetyl-CoA and malonyl-CoA. FASN overactivity has been implicated in cancer onset and progression in many cancers [65]. Interestingly, FASN has also been shown to be a transcriptional target for TRIB1 [66]. Among the specific TRIB1 interactors histone deacetylase 6 (HDAC6) stands out. HDAC6, which is mostly cytoplasmatic, has been implicated in cancer and metastasis formation in breast cancer [67]. As described above for HEK293T cells, TRIB3 interacted with a number of transcription factors that are mostly associated with transcriptional repression. The interaction with ZBTB1 was also detected in MCF7 cells among other zinc finger proteins (ZNF746, ZNF12, ZNF24) (Figure 3E), many of which are related to transcriptional repression [68]. In fact, ZBTB1 has been recently associated with resistance to tamoxifen and aerobic glycolysis in breast cancer cells [69]. Both resistance to drug treatment and glucose metabolism are major cellular pathways in which TRIB3 has been implicated before [70,71]; future studies are needed to establish whether the TRIB3–ZBTB1 interaction plays a role in these pathways. Similar to HEK293T cells, the interaction between TRIB3 and the TIM–TOM complex was also detected in MCF7 cells (Appendix A), suggesting a mitochondrial pool and function for this protein. To further validate the role of TRIB3 as a transcriptional repressor we tested the Gal4DBD-TRIB3 fusion protein, as described above, for HEK293T cells, and also detected reduced transcriptional activity in these MCF7 breast cancer cells (Figure 3F). These findings indicate that our AP–MSMS approach provides a powerful tool to unravel novel pseudokinase biology, one that is not limited to a single cell system.

## 4. Discussion

Pseudokinases, such as the three human TRIB proteins, hold promise as biomarkers in cancer, but their molecular functions are still incompletely understood. Here we reported a systematic characterization of TRIB1, -2 and -3 interactomes in HEK 293T cells to provide a better understanding of the differences and redundancies in Tribbles’ functions. In addition, our mass spectrometry-based approach revealed the importance of the intrinsically disordered N-Terminal domain of TRIB3 in the interaction with transcriptional regulatory proteins. We showed that TRIB3 is associated with transcriptional repression and that this role is mostly carried by the N-Terminal of TRIB3. Moreover, we discover new interactors of TRIB1 and -3 in breast cancer cells that might help to understand the role of these proteins in cancer pathophysiology. 

The study of the function of pseudoenzymes presents obvious difficulties in comparison with their enzymatically active counter partners, as no catalytic product can be measured as a read-out of their activity. Most of these pseudoenzymes rely on protein–protein interactions (PPI) to exert their function and several mass spectrometry-based techniques can be used for the identification of interactors, such as proximity ligation or crosslinking mass spectrometry, all with particular advantages and disadvantages [32]. Our data shows how powerful is the use of AP–MS approaches for the discovery of new interactors and the study of pseudoenzyme function. Modern mass spectrometers have a tremendous sensitivity that allows them to detect the smallest contaminant and, therefore, a quantitative filter must be introduced to differentiate between genuine interactors and background noise. These quantitative filters can be introduced in the form of isotopes or in the form of algorithms for label-free quantification; an example of the last is the intensity-based absolute quantification (iBAQ) used in this study, which allowed us to determine protein abundance. Taken together, this shows that quantitative MS-based proteomics is the most powerful method for identifying PPI and studying pseudoenzyme function to date.

Through AP–MS we confirmed previously reported Tribbles-interacting proteins and identified novel partners. As demonstrated before, we show that all three human tribbles family members can interact with the E3 ubiquitin ligase COP1. However, it seems that TRIB1 function is more dominated by the interaction with COP1, and that explains the high amount of proteasomal regulatory proteins as well as the low abundance of other interactors. TRIB2 and -3 also interact with COP1 but, next to these, many other interactors, not related to proteasomal degradation, were detected. This also seems the case when we compared the interactomes of TRIB1 and -3 in breast cancer cells. Moreover, TRIB1 protein expression was lower when compared to TRIB3 upon induction with doxycycline (Figure 3B), and, thus, could be reverted when proteasomal degradation was inhibited (data not shown), indicating that the lower amount of TRIB1 protein was the result of proteasomal degradation and was not due to different responses to doxycycline induction. TRIB1 subcellular localization appeared to be mostly cytoplasmatic in comparison with TRIB3, which showed predominant nuclear localization; this can also explain the difference in protein stability and interactomes. In addition, our data also demonstrates some interactions that had been suggested in literature before but not experimentally demonstrated, such as the interaction with p53 or the interaction with CDKN1 [72,73]. These interactions might be related to the ability of Tribbles to regulate the cell cycle and therefore the implications for cancer research are potentially very important. Both of these proteins are among the most commonly found mutated across all cancer types [74,75]. These interactions, together with others described above, might suggest a role of Tribbles in DNA damage. Furthermore, whether for example the TRIB3–CDKN1 interaction contributes to the increased proliferation observed upon long-term overexpression of TRIB3 in MCF7 cells [62] remains to be established.

Finally, we report many novel interacting proteins that interact with one or more Tribbles family members. Amongst the cellular proteins with well-established functions is the metabolic enzyme FASN. Interestingly, FASN is a major regulator of neoplastic lipogenesis and is commonly found overexpressed in many cancers [65], is a metabolic oncogene that has been suggested as an attractive target for cancer therapy [76] and, given the ability of TRIB1 and -3 to mark proteins for proteasomal degradation, this interaction represents a promising therapeutic approach for breast cancer. The class of well-characterized TRIB3 interacting proteins also includes the transcriptional repressors ZBTB1 and SPEN, which may be responsible for the transcriptional repression observed when TRIB3 is tethered to DNA. The ability of tribbles to regulate the functions of transcription factors has been reported before [29,30,31]; however, their role as a transcriptional repressor has not been shown before. While novel Tribbles interacting proteins with well-characterized functions may present immediate new entries for future research, interacting proteins with poorly understood functions such as the zinc finger proteins found interacting with TRIB3, obviously will need more characterization before their value—be it therapeutic or more fundamental—can be assessed. It should be noted that the protein kinase AKT/PKB, a previously described interaction partner of TRIB3 [14], was detected in the TRIB1 interactome but was not a dominant hit in the TRIB3 interactomes in either HEK293T or MCF7 cells. Furthermore, when we compared the TRIB3 interactomes between the genetic variants R84 and Q84—harboring an arginine and glutamine residue at position 84, respectively—no significant differences in interactomes were found (data not shown). The R variant was reported to be a more potent inhibitor of insulin signaling through stronger AKT binding when tested in hepatocytes [77]. Together, these findings support the view that TRIB interactomes may harbor a uniform component (overlap between for example HEK293T cells and MCF7 cells) as well as a flexible component that depends on cell type (e.g., hepatocyte vs. HEK293T cells vs. MCF7 breast cancer cells) or cellular status (e.g., proliferative status, metabolic status). Analyzing and comparing TRIB interactomes in more cell types and under different conditions is, therefore, an important future direction. 

In summary, we have shown new interactions that might be very relevant for cancer therapy and could situate Tribbles as therapeutic targets in breast cancer and we have shown how powerful and useful is the study of tribbles’ functions through MS–based proteomics approaches.

## 5. Conclusions

We have used an MS–based approach to find new interactors of Tribbles proteins that might serve as starting point for future research. We have shown the ability of TRIB3 to function as a transcriptional repressor as we looked at the similarities and differences between TRIB1 and -3 in breast cancer cells, finding new interactors that might help to understand better the function of these proteins in breast cancer pathology.

## Figures and Tables

**Figure 1 cancers-13-06318-f001:**
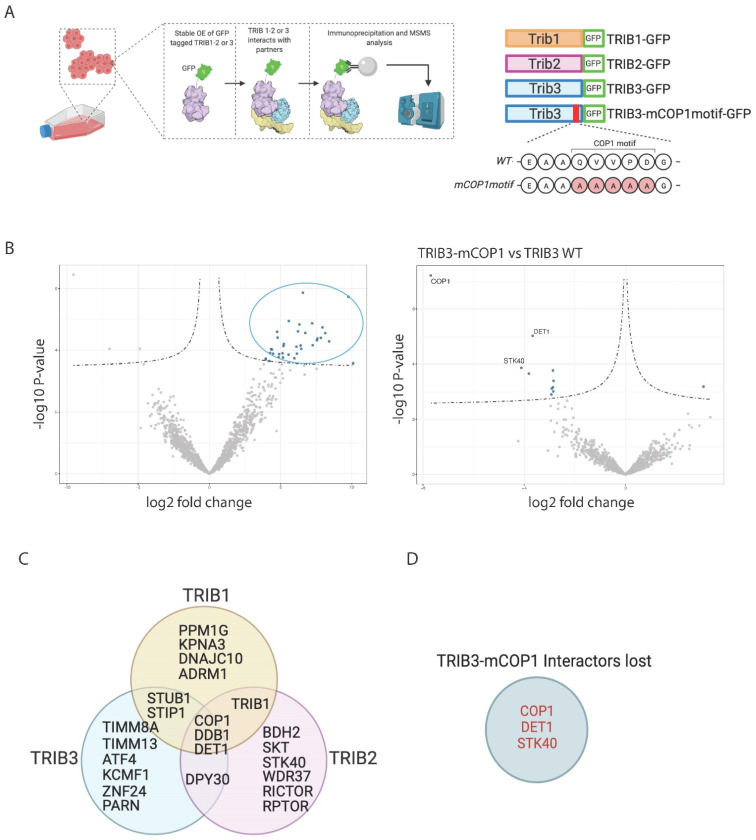
TRIB1, -2 and -3 interactomes in HEK293T cells. (**A**) Schematic representation of workflow followed for AP-MS experiments. Figure was created with BioRender.com. (**B**) Volcano plots showing TRIB3 interactors compare to GFP control and TRIB3-mCOP1 interactors compared to WT TRIB3 in HEK293T cells. (**C**) Venn diagram of common and unique interactors between Tribbles family members. (**D**) TRIB3 interactors lost when comparing TRIB3 vs TRIB3-mCOP1 interactomes.

**Figure 2 cancers-13-06318-f002:**
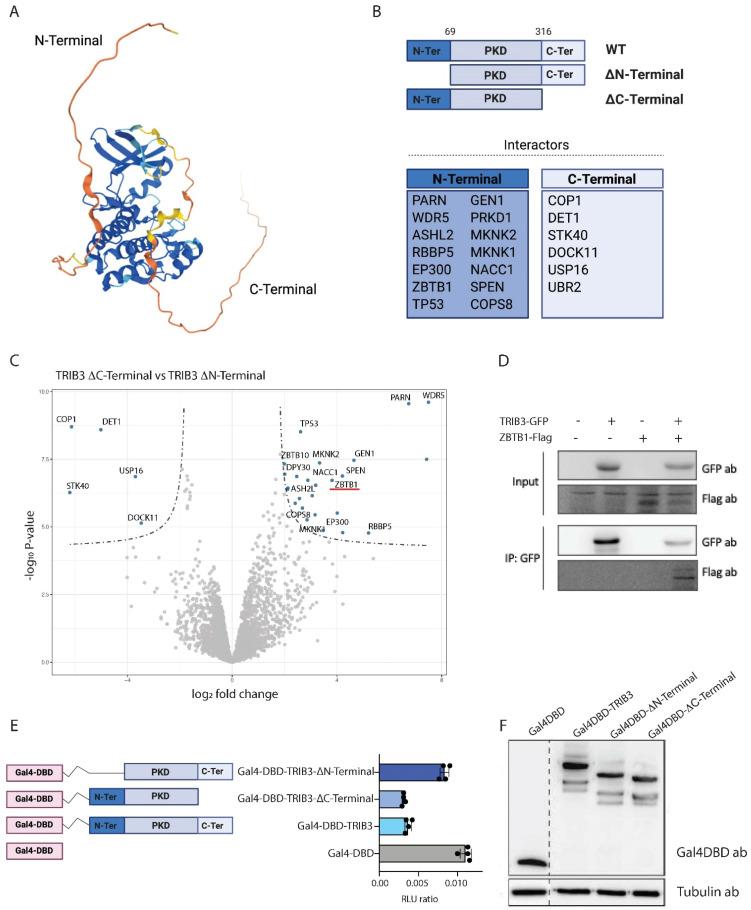
Contribution of TRIB3 domains to its interactome and function. (**A**) Human TRIB3 structure prediction by AlphaFold. Colors represent pLDDT score (blue: very high confidence, light blue: confident, yellow: low and orange: very low or unstructured). (**B**) Schematic representation of TRIB3 mutants and specific interactors lost when the indicated domain was removed. (**C**) Volcano plot showing interaction of the N-terminal and C-terminal domains of TRIB3 in HEK293T cells. (**D**) Co-immunoprecipitation assay of TRIB3-GFP and ZBTB1-Flag in HEK293T cells. (**E**) Gal4 reporter assay of Gal4-DBD, Gal4-TRIB3, Gal4-TRIB3-ΔN-terminal and Gal4-TRIB3-ΔC-terminal in Hek293T cells. Data is normalized using Renilla luciferase. (**F**) Western blot using Gal4DBD and tubulin antibodies showing similar expression of the constructs used for the Gal4 reporter assay.

**Figure 3 cancers-13-06318-f003:**
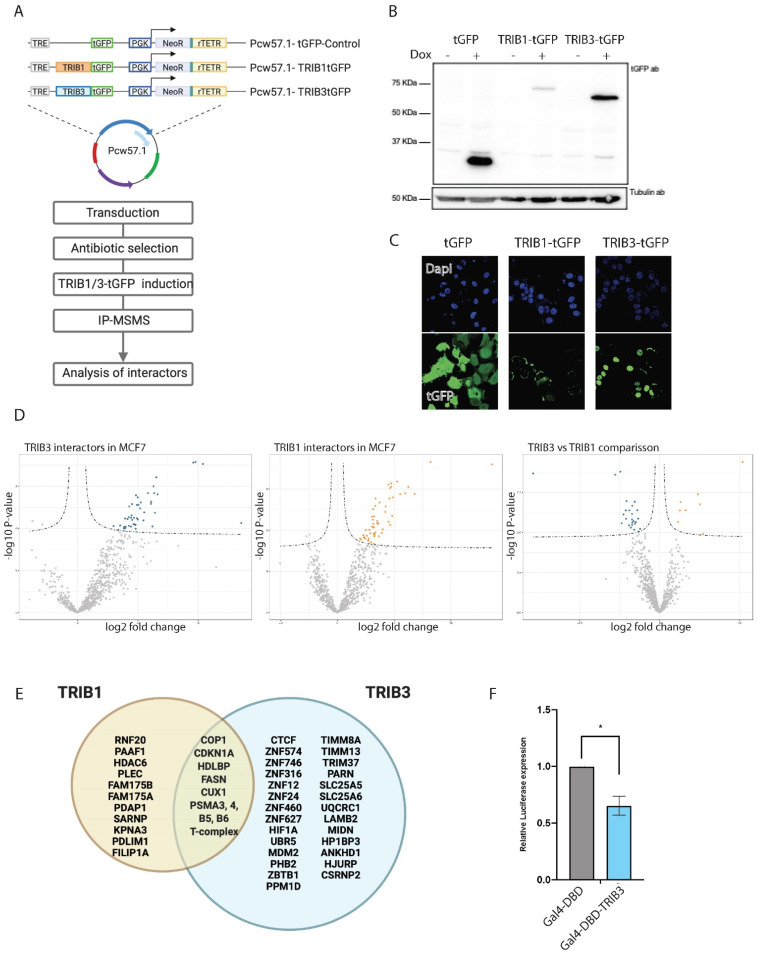
TRIB1 and TRIB3 interactors in MCF7 cells. (**A**) Schematic representation of inducible constructs and workflow of the AP–MS experiments followed in MCF7 cells. (**B**) Western blot using t-GFP antibody sowing inducible expression of TRIB1-tGFP and TRIB3-tGFP upon doxycycline treatment and Tubulin expression as loading control. (**C**) Confocal images taken at 40× magnification showing tGFP, TRIB1-tGFP and TRIB3-tGFP localization upon doxycycline induction. (**D**) Volcano plots of TRIB1 and TRIB3 interactors compared with tGFP control in MCF7 cells and a volcano plot showing the comparison between TRIB1 and TRIB3 interactome in these cells. (**E**) Venn diagram of similar and different interactors between TRIB1 and TRIB3 in MCF7 cells detected in the AP–MSMS experiments. (**F**) Gal4 reporter assay of Gal4-DBD and Gal4-TRIB3 in MCF7 cells. Data is normalized using Renilla luciferase. Data is indicated as mean ± SEM. *p*-values were calculated using two-tailed Student’s *t*-test (* *p* < 0.05).

**Table 1 cancers-13-06318-t001:** Top 25 TRIB1 interacting partners in HEK293T cells based on *p*-value.

Gene Name	-Log (*p*-Value)	Adjusted *p*-Value	Log2 Fold Change *	Full Name
*DNAJC10*	6.4354	6.18 × 10^−5^	7.300	DnaJ Heat Shock Protein Family Member C10
*DDB1*	5.8200	0.04	4.678	Damage Specific DNA Binding Protein 1
*TRIB1*	5.7117	8.17 × 10^−7^	2.567	Tribbles Pseudokinase 1
*PPM1G*	5.2926	0.02	6.113	Protein Phosphatase Mg^2+^ Dependent 1G
*PSMA1*	5.2130	5.29 × 10^−5^	4.699	Proteosome 20S subunit Alpha 1
*PSMA2*	5.0433	0.0002	3.443	Proteosome 20S subunit Alpha 2
*PSMD4*	5.0242	0.0004	3.112	Proteasome 26S Subunit Ubiquitin receptor 4
*HDAC6*	4.9461	0.03	4.223	Histone Deacetylase 6
*PSMC6*	4.5066	3.12 × 10^−5^	4.667	Proteosome 26S subunit ATPase 6
*ADRM1*	4.4064	0.0004	5.600	26S Proteosome Ubiquitin Receptor
*PSMC2*	4.3112	4.68 × 10^−5^	2.554	Proteosome 26S subunit ATPase 2
*PSMB6*	4.1668	0.006	6.433	Proteosome 20S subunit Beta 7
*STIP1*	3.9432	0.01	4.001	Stress induced Phosphoprotein 1
*PSMB2*	3.9126	1.70 × 10^−5^	3.333	Proteosome 20S subunit Beta 2
*STUB1*	3.8218	0.05	5.677	STIP1 Homology and U-Box Containing Protein 1
*PSMC1*	3.7980	9.63 × 10^−6^	3.655	Proteosome 20S subunit Beta 7
*HSPH1*	3.6860	0.004	5.911	Heat Shock Protein Family H Member 1
*KPNA4*	3.6243	0.001	3.770	Karyopherin Subunit Alpha 4
*DET1*	3.5614	5.29 × 10^−5^	3.190	DET 1 Partner of COP1 E3 Ubiquitin Ligase
*HSPA4L*	3.4984	0.003	4.675	Heat Shock Protein Family A Member 4 Like
*RFWD2*	3.3768	0.0004	9.453	COP1 E3 Ubiquitin Ligase
*PSMA5*	3.3478	0.0004	2.724	Proteosome 20S subunit Alpha 5
*HSPA4*	3.0482	1.30 × 10^−5^	5.119	Heat Shock Protein A Member 4
*PSMA7*	3.0482	0.003	3.880	Proteosome 20S subunit Alpha 7
*KPNA3*	2.7785	0.001	4.654	Karyopherin Subunit Alpha 3

***** Log 2-fold change calculated using mean intensity of TRIB1 condition compared to GFP.

**Table 2 cancers-13-06318-t002:** Top 25 TRIB22 interacting partners in HEK293T cells based on *p*-value.

Gene Name	-Log (*p*-Value)	Adjusted *p*-Value	Log2 Fold Change *	Full Name
*TRIB1*	5.771	2.76 × 10^−5^	12.364	Tribbles Pseudokinase 1
*USP11*	5.541	0.001	4.356	Ubiquitin Specific Peptidase 11
*ISCA1*	5.391	0.0002	6.115	IRON-Sulfur Cluster Assembly 1
*BDH2*	4.951	3.90 × 10^−6^	8.657	3-Hydroxybutyrate Dehtdrogenase 2
*ZKSCAN1*	4.945	0.0001	6.503	Zinc Finger with KRAB and SCAN Domains 1
*DET1*	4.885	3.4 × 10^−5^	7.812	DET 1 Partner of COP1 E3 Ubiquitin Ligase
*DDB1*	4.523	0.001	4.456	Damage Specific DNA Binding Protein 1
*AIFM1*	4.274	3.27 × 10^−5^	4.898	Apoptosis inducing Factor Mitochondrial Associated 1
*FECH*	4.262	2.76 × 10^−5^	5.878	Ferrochelatase
*WDR37*	4.007	0.001	7.058	WD Repeat Domain 37
*KIAA1217*	3.789	0.001	9.837	Sickle tail Protein Homolog
*KCTD21*	3.751	0.002	3.573	BTB/POZ Domain-Containing Protein KCTD21
*CDC42EP1*	3.686	0.0003	3.082	CDC42 Effector Protein 1
*MLF2*	3.471	0.0003	4.887	Myeloid Leukemia factor 1
*STK40*	3.231	0.0001	7.644	Serine/Threonine Kinase 40
*RAB3GAP1*	3.218	0.0002	3.543	RAB3 GTPase Activating Protein Catalytic Subunit 1
*EMD*	3.153	0.01	4.316	Emerin
*RFWD2*	3.034	0.002	10.620	COP1 E3 Ubiquitin Ligase
*RPTOR*	2.864	0.002	2.346	Regulatory Associated Protein of MTOR Complex 1
*FKBP4*	2.841	0.01	4.293	FKBP Prolyl Isomerase 4
*SRCIN1*	2.762	0.0004	4.617	SRC Kinase Signaling Inhibitor 1
*PKP2*	2.646	0.0004	4.004	Plakophilin 2
*TBC1D4*	2.634	0.001	2.375	TBC1 Domain Family Member 4
*HAUS8*	2.163	0.002	3.115	HAUS Augmin Like Complex Subunit 8
*RICTOR*	1.999	0.001	2.706	RPTOR Independent Companion of MTOR Complex 2

* Log 2-fold change calculated using mean intensity of TRIB2 condition compared to GFP.

**Table 3 cancers-13-06318-t003:** Top 25 TRIB3 interacting partners in HEK293T cells based on *p*-value.

Gene Name	-Log (*p*-Value)	Adjusted *p*-Value	Log2 Fold Change *	Full Name
*TIMM13*	7.1286	0.0002	12.187	Translocase of Inner Mitochondrial Membrane 13
*TRIM37*	6.9472	0.006	6.961	Tripartite Motif Containing 37
*DDB1*	6.2442	0.005	5.187	Damage Specific DNA Binding Protein 1
*KCMF1*	5.7442	0.007	5.885	Potassium Channel Modulatory Factor 1
*ATF4*	5.7130	0.01	1.345	Activating Transcription factor 4
*STUB1*	5.4533	0.03	5.972	STIP1 Homology and U-Box Containing Protein 1
*MLLT11*	5.0745	0.002	6.917	MLLT11 Transcription factor 7 Cofactor
*RFWD2*	5.0033	0.0006	2.489	COP1 E3 Ubiquitin Ligase
*PARN*	4.9405	0.01	7.321	Poly(A)-Specific Ribonuclease
*PRKD2*	4.6601	0.001	7.015	Protein kinase D2
*PASK*	4.6442	0.005	8.091	PAS Domain Containing Serine/Threonine Kinase
*DPY30*	4.3027	0.04	5.462	Dpy-30 Histone Methyltransferase Complex
*DET1*	4.1258	0.04	5.134	DET 1 Partner of COP1 E3 Ubiquitin Ligase
*ZNF24*	4.0712	0.001	4.125	Zinc Finger Protein 24
*EP300*	3.9599	0.01	6.900	E1A Binding Protein P300
*STIP1*	3.9000	0.020.	3.582	Stress induced Phosphoprotein 1
*KANK2*	3.7899	0.16	6.740	KN Motif Ankyrin Repeat Domains 2
*RBBP8*	3.7371	0.019	7.875	RB Binding Protein 8
*ZNF507*	3.6606	0.006	7.112	Zinc Finger Protein 507
*PPP6C*	3.5073	0.03	7.964	Protein Phosphatase 6 Catalytic Subunit
*WDR62*	3.5072	0.006	7.074	WD Repeat Domain 62
*PPP6R3*	3.4224	0.003	7.112	Protein Phosphatase 6 Regulatory Subunit 3
*AKAP8L*	2.7110	0.0005	8.183	A-Kinase Anchoring Protein 8 Like
*ZNF655*	2.6525	0.01	6.810	Zinc Finger Protein 655
*TIMM8A*	2.6021	3.48 × 10^−5^	12.262	Translocase of Inner Mitochondrial Membrane 8A

* Log 2-fold change calculated using mean intensity of TRIB3 condition compared to GFP.

**Table 4 cancers-13-06318-t004:** Top binding partners of TRIB3 ΔN- and ΔC-Terminal domains based on *p*-value.

Gene Name	-Log (*p*-Value)	Adjusted *p*-Value	Log2 Fold Change *	Full Name
	TRIB3-ΔC-Terminal Binding partners			
*RFWD2*	4.4826	4.54 × 10^−6^	5.943	COP1 E3 Ubiquitin Ligase
*STK40*	3.1119	0.0002	6.370	Serine/Threonine Kinase 40
*DOCK11*	3.7967	0.004	3.498	Dedicator of Cytokinases 11
*DET1*	3.8995	1.02 × 10^−5^	4.667	DET 1 Partner of COP1 E3 Ubiquitin Ligase
*USP16*	2.5863	0.01	3.822	Ubiquitin Specific Peptidase 16
*UBR2*	3.1640	0.001	2.610	Ubiquitin Protein Ligase E3 Component N-Recognin 2
*DDI2*	1.8154	0.02	1.811	DNA Damage 1 Homolog 2
	TRIB3-ΔN-Terminal Binding partners			
*MKNK2*	6.0395	0.007	3.525	MAPK Interacting Serine/Threonine Kinase 2
*WDR5*	5.7443	4.38 × 10^−7^	7.841	WD Repeat Domain 5
*COPS8*	5.6291	0.01	3.123	COP9 Signalosome Subunit 8
*TP53*	5.3723	3.16 × 10^−6^	2.800	Tumor Protein P53
*PARN*	5.0650	4.38 × 10^−7^	6.298	Poly(A)-Specific Ribonuclease
*GEN1*	4.8329	1.35 × 10^−5^	4.438	Gen1 Holliday Junction 5′ Flap Endonuclease
*SPEN*	3.8986	3.87 × 10^−6^	4.154	SPEN Family Transcriptional Repressor
*ZNFP24*	3.8823	0.0001	1.719	Zinc Finger Protein 91
*NACC1*	3.7582	3.08 × 10^−5^	2.193	Nucleus Accumbens Associated 1
*ZBTB1*	3.5994	8.91 × 10^−5^	4.004	Zinc Finger And BTB Domain 1
*SETD2*	3.5315	0.002	1.627	SET Domain 2 Histone Lysine Methyltranferase
*DPY30*	3.4148	5.56 × 10^−5^	2.296	Dpy-30 Histone Methyltranferase Complex Subunit
*RBBP5*	3.2480	0.0009	4.471	RB Binding Protein 5
*MYC*	2.6206	0.01	2.233	MYC Proto-Oncogene
*EP300*	2.5832	0.005	4.017	E1A Binding Protein P300
*MKNK1*	2.5381	0.05	3.411	MAPK Interacting Serine/Threonine Kinase 1
*ASH2L*	2.4872	0.001	5.577	Set1/Ash2 Histone Methyltranferase Complex Subunit
*AKAP1*	2.2902	0.005	2.712	A-Kinase Anchoring Protein 1
*PRKD1*	1.7714	0.03	4.106	Protein kinase D1
*NFAT4*	3.936	0.008	2.464	Nuclear Factor of Activated T Cells 3

* Absolute Log 2-fold change calculated using mean intensity of ΔN-Terminal condition compared to ΔC-Terminal condition.

**Table 5 cancers-13-06318-t005:** Top 20 TRIB1 and TRIB3 binding partners in MCF7 cells based on *p*-value.

Gene Name	-Log (*p*-Value)	Adjusted *p*-Value	Log2 Fold Change	Full Name
	TRIB1 binding partners in MCF7			
*PFN1*	6.8434	0.007	2.550	Profilin 1
*CDKN1A*	6.7563	0.0002	2.315	Cyclin Dependent Kinase Inhibitor 1A
*CCT5*	6.0523	0.0002	2.083	T-complex protein 1 subunit epsilon
*PDAP1*	6.0427	0.006	2.151	PDGFA Associated Protein 1
*PRKDC*	5.8532	0.0001	2.508	DNA-dependent protein kinase catalytic subunit
*ERH*	5.8170	4.16 × 10^−5^	3.553	ERH MRNA Splicing and Mitosis Factor
*RFWD2*	5.5455	1.08 × 10^−8^	6.240	COP1 E3 Ubiquitin Ligase
*RNF40*	5.3754	7.73 × 10^−6^	3.083	E3 ubiquitin-protein ligase BRE1B
*FASN*	5.3652	0.01	1.708	Fatty Acid Synthase
*MTHFD1*	5.0852	4.23 × 10^−5^	2.525	C-1-tetrahydrofolate synthase
*STIP1*	4.9889	2.89 × 10^−5^	3.699	Stress-induced-phosphoprotein 1
*DNAJB1*	4.9816	2.18 × 10^−7^	5.567	DnaJ Heat Shock Protein Family Member B1
*STK40*	4.9431	5.94 × 10^−6^	5.105	Serine/threonine Kinase 40
*PLEC*	4.3460	0.0006	3.183	Plectin
*HDAC6*	4.1498	0.01	2.356	Histone Deacetylase 6
*PAAF1*	3.9612	0.008	2.609	Proteasomal ATPase Associated Factor 1
*EDF1*	3.8339	1.82 × 10^−5^	3.326	Endothelial Differentiation Related Factor 1
*CUX1*	3.7878	0.003	2.122	Cut like Homeobox 1
*PRDX2*	3.7664	0.01	3.741	Peroxiredoxin 2
*CEBPB*	2.9757	0.01	1.223	CCAAT Enhancer Binding Protein Beta
	TRIB3 binding partners in MCF7			
*CDKN1A*	7.7691	0.0001	2.752	Cyclin Dependent Kinase Inhibitor 1A
*TRIM37*	7.6337	0.01	3.405	Tripartite Motif Containing 37
*ZNF217*	7.5252	2.35 × 10^−6^	2.732	Zinc Finger Protein 217
*TRIB1*	6.5117	0.001	2.966	Tribbles Pseudokinase 1
*HIF1A*	6.4611	0.0001	2.139	Hypoxia Inducible Factor 1 Subunit Alpha
*PPM1D*	6.4143	0.006	2.567	Protein Phosphatase Mg Dependent 1D
*ZBTB1*	6.0395	0.01	1.828	Zinc Finger And BTB Domain 1
*CEBPB*	5.7224	0.0001	2.463	CCAAT/enhancer-binding protein beta
*WDR5*	5.7100	0.001	2.372	WD repeat-containing protein 5
*ZNF627*	5.3262	0.003	2.866	Zinc Finger Protein 627
*ZNF460*	5.1982	0.002	1.775	Zinc Finger Protein 460
*DNAJB1*	5.1152	2.35 × 10^−6^	4.064	DnaJ homolog subfamily B member 1
*PARN*	4.9613	2.48 × 10^−5^	3.301	Poly(A)-specific Ribonuclease
*RFWD2*	4.9105	5.62 × 10^−7^	3.904	COP1 E3 Ubiquitin Ligase
*FASN*	4.6569	0.05	1.709	Fatty Acid Synthase
*TIMM13*	3.8986	1.92 × 10^−5^	2.370	Translocase of Inner Mitochondrial Membrane 13
*ZNF12*	3.7582	0.01	2.602	Zinc Finger Protein 12
*KDM3B*	3.5315	0.05	2.185	Lysine-specific demethylase 3B
*DPY30*	3.3126	0.001	2.089	Dpy-30 Histone Methyltranferase Complex
*TP53*	2.3725	0.0001	2.372	Cellular tumor antigen p53

* Absolute log 2-fold change calculated using mean intensity of TRIB3 condition compared to TRIB1 condition.

## Data Availability

Data are available via ProteomeXchange with identifier PXD030404.

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
