# Peer review of "Comprehensive Profiling of Mammalian Tribbles Interactomes Implicates TRIB3 in Gene Repression"

_cancers, 2021, doi:10.3390/cancers13246318_

Round 1
Reviewer 1 Report
The study by Hernández-Quiles sets out to map the interactome of the three Tribbles proteins. The premise of the study is relevant, with the approach of testing interacting partners of N- and C-terminal truncations interesting. If improved I think would be interesting to the field. However, both in execution and in presentation I believe the data falls short of being publishable in it’s current form. The treatment of the proteomic data is not clear, and experiments validating the role of TRIB3 in gene repression have no clear link due to the nature of the assay used (at least as explained). Also the presentation of the data needs to be improved.
Main Issues
-It is unclear from the methods sections if multiple testing correction has been performed? The -Log (p-value) in the table is normally only appropriate for visualising the data, whereas in a table like this the adjusted p-value after correcting for multiple comparisons should be reported.
-The tables do not appear to be in any specific order. The hits do not appear in order of name, p-value, or Fold change, which does not give a clear representation of the data. Also I presume the Fold-change reported in the tables is a Log2 fold change? (It is difficult to compare to the volcano plots due to such small labels, but these should clearly indicate what the negative and positive fold change is in relation to (rather than interpreting from a non-descript title). Thresholds used to include data in the table should also be included.
-The rationale is not clear for performing a plasmid-based reporter assay, based on the finding that Tribbles interact with components of the transcriptional machinery. Are the authors suggesting that TRIB3 directly modulates transcription, or that specific proteins (presumably ZBTB1 based on the organisation) implicated by the proteomic data are recruited to the plasmid regulate expression? Are ZBTB1 and/SPEN supposed to be working with this plasmid? My impression is that repressors work through a range or regulatory mechanisms, chromatin structures, epigenetic modifiers, so I would be surprised to find native TRIB3 partners recruited to a transected plasmid and performing their role. Without additional data there are too many missing links in this mechanism and these assays is very artificial.
-Why was the human cell line data analysis performed against a mouse proteome library?
-The figures, and especially the labels throughout are too small to be legible and should be clarified
-The localisation of TRIB1 GFP is proposed to be cytoplasmic, but as the authors themselves indicate there is significant overlap in interactions with TRIB3 (which is nuclear). Can the authors comment on this, and also the cytoplasmic localisation of TRIB1 relative to the work of Kung and Jura (EMBO 2019) where TRIB1 is mostly nuclear.
-The reference section uses a reference format that is very hard to follow (authors all as initials).
Minor
-There are some minor editing issues, inconsistent capitalisation, and typographical errors that should be tidied up
-In the introduction pseudokinases are referred to as resembling serine/threonine kinases, however there are a good number that resemble tyrosine kinases also
-Several parts of the methods appear to have errors in units (I think from missing µ symbol), For instance 2 g/mL doxycycline seems like a lot!
-Figure 1C is referred to on Page 5,in relation to an experiment mutating the COP1 binding motif. Figure 1C is a venn diagram showing interactions of the three Tribbles proteins, and does not show data for the COP1 mutant TRIB3.
Author Response
Point-by-point response to Reviewer #1
First and foremost, we would like to thank the reviewer for taking the time to review our manuscript and we appreciate the positive and constructive comments and suggestions. We were pleased to find that the reviewer indicated that our study ‘if improved, would be interesting to the field’. We have now adapted the manuscripts in light of the reviewer’s comments. Please find below a detailed point-by-point description of the changes we made. We hope that these revisions will make our manuscript suitable for publication in ‘Cancers’.
Reviewer 1:
“It is unclear from the methods sections if multiple testing correction has been performed? The -Log (p-value) in the table is normally only appropriate for visualising the data, whereas in a table like this the adjusted p-value after correcting for multiple comparisons should be reported.”
We apologize for this omission: multiple testing correction was in fact performed earlier but not mentioned explicitly and we have now added adjusted p-values to the tables.
“The tables do not appear to be in any specific order. The hits do not appear in order of name, p-value, or Fold change, which does not give a clear representation of the data. Also I presume the Fold-change reported in the tables is a Log2 fold change? (It is difficult to compare to the volcano plots due to such small labels, but these should clearly indicate what the negative and positive fold change is in relation to (rather than interpreting from a non-descript title). Thresholds used to include data in the table should also be included.”
We agree that the tables were confusing. Hits have now been ordered according to their p-value and tables and figures legends have been clarified. In addition, the inclusion criteria is explained in the methods section.
“The rationale is not clear for performing a plasmid-based reporter assay, based on the finding that Tribbles interact with components of the transcriptional machinery. Are the authors suggesting that TRIB3 directly modulates transcription, or that specific proteins (presumably ZBTB1 based on the organisation) implicated by the proteomic data are recruited to the plasmid regulate expression? Are ZBTB1 and/SPEN supposed to be working with this plasmid? My impression is that repressors work through a range or regulatory mechanisms, chromatin structures, epigenetic modifiers, so I would be surprised to find native TRIB3 partners recruited to a transected plasmid and performing their role. Without additional data there are too many missing links in this mechanism and these assays is very artificial.”
The plasmid-based assays in our study were intended to obtain a first view of the potential transcriptional effects of TRIB3. While TRIB3 has been shown to interact with several transcription factors (e.g. C/EBPalpha, PPARgamma, ATF4), its exact role in gene regulation is unclear as this pseudokinase lacks enzymatic activity (towards transcription factors or for example, histone proteins). We hypothesized therefore that TRIB3 may activate or repress transcription by recruiting (epigenetic) gene regulators and indeed observed several of these proteins in the TRIB3 interactomes in HEK293T and MCF7 cells. As we detected both (epigenetic) activators and repressors, we artificially tethered TRIB3 to the DNA (Gal4DBD fusion protein) and assayed transcriptional activity (5xGal4 reporter). This type of approach has been successfully used to classify for example ZBTB1 (1) and TRIM28 (2) (two of the TRIB3 interacting proteins identified here) but also epigenetic proteins like histone deacetylases (HDACs; (3)) as transcriptional repressors. We observed transcriptional repression, suggesting that TRIB3 predominantly recruits transcriptional repressors under these experimental conditions. In addition, repressive activity is lost upon deletion of the N-terminus, the domain where repressors (and activators) bind. We realize now that we may have unintentionally given the impression that the repressive activity was due to the interaction with ZBTB1, but this conclusion cannot be drawn at this moment and we have adapted the Abstract and the Discussion accordingly. We feel that in-depth studies on the role of TRIB3 in the context of chromatin are beyond the scope of the current work, studies that will also be hampered by the poor quality of commercially available TRIB3 antibodies (our unpublished observations).
“Why was the human cell line data analysis performed against a mouse proteome library?”
We sincerely thank the reviewer for pointing this out: the data was analyzed against a human proteome library but wrongly indicated due to an unfortunate error on our side. This is now corrected in the Materials and Methods section.
“The figures, and especially the labels throughout are too small to be legible and should be clarified”
We have adjusted figures and legends; all figures were checked against the journal's guidelines.
-“The localisation of TRIB1 GFP is proposed to be cytoplasmic, but as the authors themselves indicate there is significant overlap in interactions with TRIB3 (which is nuclear). Can the authors comment on this, and also the cytoplasmic localisation of TRIB1 relative to the work of Kung and Jura (EMBO 2019) where TRIB1 is mostly nuclear.”
As pointed out in our Introduction, different and even contradictory observations have been made regarding the subcellular localization of Tribbles proteins. As the reviewer indicates, Kung and Jura (2019) showed predominantly nuclear localization of TRIB1 when overexpressed in HEK293T cells. We made similar observations (data not shown) but observed predominant cytoplasmatic localization of TRIB1 in MCF7 cells (Figure 3). TRIB3 was mainly (but not exclusive) nuclear in HEK293T and MCF7 cells. Together with literature data, our observations indicate that the localization of Tribbles proteins depends on cellular context and conditions. Also, subcellular localization can for example be predominantly nuclear but not exclusively nuclear. We now refer to the Kung and Jura paper in our Introduction and expanded the text on this point.
“The reference section uses a reference format that is very hard to follow (authors all as initials).”
We understand the comment of the reviewer and we agree that it may be hard to follow sometimes. We have used the format indicated by the journal in the author's guidelines.
“There are some minor editing issues, inconsistent capitalisation, and typographical errors that should be tidied up”
We apologize for this and have corrected typos and gene/protein nomenclature mistakes throughout the manuscript. We thank the reviewer for his/her careful review.
“Figure 1C is referred to on Page 5, in relation to an experiment mutating the COP1 binding motif. Figure 1C is a venn diagram showing interactions of the three Tribbles proteins, and does not show data for the COP1 mutant TRIB3.”
A Cartoon showing the most significant hits lost when the interactomes of TRIB3 versus TRIB3mCOP1 are compared has been added to Figure 1.
References:
- Matic I, Schimmel J, Hendriks IA, van Santen MA, van de Rijke F, van Dam H, Gnad F, Mann M, Vertegaal AC. Site-specific identification of SUMO-2 targets in cells reveals an inverted SUMOylation motif and a hydrophobic cluster SUMOylation motif. Mol Cell. 2010 Aug 27;39(4):641-52. doi: 10.1016/j.molcel.2010.07.026. PMID: 20797634.
- Moosmann P, Georgiev O, Le Douarin B, Bourquin JP, Schaffner W. Transcriptional repression by RING finger protein TIF1 beta that interacts with the KRAB repressor domain of KOX1. Nucleic Acids Res. 1996 Dec 15;24(24):4859-67. doi: 10.1093/nar/24.24.4859. PMID: 9016654; PMCID: PMC146346.
- Yang WM, Yao YL, Sun JM, Davie JR, Seto E. Isolation and characterization of cDNAs corresponding to an additional member of the human histone deacetylase gene family. J Biol Chem. 1997 Oct 31;272(44):28001-7. doi: 10.1074/jbc.272.44.28001. PMID: 9346952.

Reviewer 2 Report
This paper was described that 3 Tribbles proteins have interactions with several distinguishing genes. Such This work such as using MS-based profiling is interesting to a wider researcher across a diverse range of cancer area and might provide an important contribution for setting several caner biomarkers in the future. I would recommend it for acceptance after some minor points described below and annotated on the manuscript are addressed.
Minor comment:
- It is recommend referring a review paper described about TRIB3 regulation and expression for your introduction or discussion page, Cancers 2021, 13, 1822, if you like.
- Three Tribbles commonly interacted 3 genes including DDB1. Could you tell us what is DDB1 and how important for DNA damage in normal cells (ex. HEK293T) or cancer cells (ex. MCF7), respectively?
- Reviewer could not find the word Figure 1B (page 5, lane214; 1B was correct instead of 1C?), 1C (page 5, lane 222; 1C was correct instead of 1A?), and 3D (page 12?) although the data was shown in these figures and figure legends.
Author Response
Point-by-point response to Reviewer #2
First and foremost, we would like to thank the reviewer for taking the time to review our manuscript and we appreciate the positive and constructive comments and suggestions. We were pleased to find that the reviewer indicated that our study “is interesting to a wider researcher across a diverse range of cancer area and might provide an important contribution for setting several cancer biomarkers in the future. I would recommend it for acceptance after some minor points described below and annotated on the manuscript are addressed.” We have now adapted the manuscript in light of the reviewer’s comments. Please find below a detailed point-by-point description of the changes we made. We hope that these revisions will make our manuscript suitable for publication in ‘Cancers’.
“It is recommend referring a review paper described about TRIB3 regulation and expression for your introduction or discussion page, Cancers 2021, 13, 1822, if you like.”
We have incorporated the reference suggested (Stefanova et al. 2021(Cancers)).
“Three Tribbles commonly interacted 3 genes including DDB1. Could you tell us what is DDB1 and how important for DNA damage in normal cells (ex. HEK293T) or cancer cells (ex. MCF7), respectively?”
We indeed identified DDB1 often as an interacting partner of TRIB1/2 and 3. DDB1 is indeed known for its role in DNA damage repair but also functions as an E3 ubiquitin adaptor protein; it has been shown in Arabidopsis that COP1, DET1, and DDB1 form a complex to enhance COP1 ubiquitin capacity of target proteins (1). We think that the interaction between TRIB proteins and DDB1 is a reflection of this same mechanism, and the role of DDB1 in this context is more as an adaptor protein to facilitate the ubiquitination of COP1 substrates. We have clarified this in the manuscript and also agree with the reviewer that a role for TRIB proteins in DNA damage cannot be excluded; this is now also indicated in the discussion section.
“Reviewer could not find the word Figure 1B (page 5, lane214; 1B was correct instead of 1C?), 1C (page 5, lane 222; 1C was correct instead of 1A?), and 3D (page 12?) although the data was shown in these figures and figure legends.”
We thank the reviewer for pointing this out and we have corrected these mistakes when referring to figures 1D and 3D (highlighted in the manuscript).
References:
- Yanagawa Y, Sullivan JA, Komatsu S, Gusmaroli G, Suzuki G, Yin J, Ishibashi T, Saijo Y, Rubio V, Kimura S, Wang J, Deng XW. Arabidopsis COP10 forms a complex with DDB1 and DET1 in vivo and enhances the activity of ubiquitin conjugating enzymes. Genes Dev. 2004 Sep 1;18(17):2172-81. doi: 10.1101/gad.1229504. PMID: 15342494; PMCID: PMC515294.

Reviewer 3 Report
The manuscript describes the characterization of binding partners of TRIB pseudokinases. Pseudokinases are especially difficult to understand functionally as they operate manly through protein binding. This is well explained by the authors and the knowledge-gap on TRIB binding partners is clearly filled by this manuscript. Although the current manuscript is largely descriptive with limited follow-up on the functional biology of newly discovered binding partners of TRIBs, the high technical quality of the experiments, the context provided on the data by the authors and need for this high quality data for the field to progress, prompts me to advice acceptance after minor revision. Publishing the data in this manuscript will provide the field handhelds for follow-up and further validation of various binding partners and facilitate rapid implementation of this manuscripts findings in diseases like cancer.
My minor revision requests are:
- Elaborate on the rationale why TRIB3 is focussed on in 3.2. Now its unclear why the study suddenly switches to scrutinizing TRIB3 instead of TRIB1/2 or all of them as in 3.1 and coming back to that line in 3.4. Also, elaborate on why studying the different domains of TRIB3 is essential, as it is unclear to me how this is required for proceeding with the new found binding partners for potential cancer therapy (which is the goal of the study according to the introduction)
- Make sure that gene/protein nomenclature is correct throughout the manuscript. E.g. TRIB3 (Human protein) and Trib3 (mouse protein) are mixed in the figures and text, but I assume all refer to human in this context. Or are mouse and human protein both used in this manuscript? Please clarify.
- In light of TP53 and CDKN1A binding, are there any effects on cell proliferation or cell death after TRIB1/2/3 expression in 293T or MCF7 cells?
- In the manuscript text comparisons are made between the 293T and MCF7 data but there is no table or venn-diagram included to facilitate this comparison. Please include a table and venn that describes the overlap between TRIB1/3 binders in 293T vs. MCF7.
Author Response
Point-by-point response to Reviewer #3
First and foremost, we would like to thank the reviewer for taking the time to review our manuscript and we appreciate the positive and constructive comments and suggestions. We were pleased to find that the reviewer indicated that “the knowledge-gap on TRIB binding partners is clearly filled by this manuscript. Although the current manuscript is largely descriptive with limited follow-up on the functional biology of newly discovered binding partners of TRIBs, the high technical quality of the experiments, the context provided on the data by the authors and need for this high quality data for the field to progress, prompts me to advice acceptance after minor revision. Publishing the data in this manuscript will provide the field handhelds for follow-up and further validation of various binding partners and facilitate rapid implementation of this manuscripts findings in diseases like cancer.” We have now adapted the manuscript in light of the reviewer’s comments. Please find below a detailed point-by-point description of the changes we made. We hope that these revisions will make our manuscript suitable for publication in ‘Cancers’.
“Elaborate on the rationale why TRIB3 is focussed on in 3.2. Now its unclear why the study suddenly switches to scrutinizing TRIB3 instead of TRIB1/2 or all of them as in 3.1 and coming back to that line in 3.4. Also, elaborate on why studying the different domains of TRIB3 is essential, as it is unclear to me how this is required for proceeding with the new found binding partners for potential cancer therapy (which is the goal of the study according to the introduction)”
We would like to thank Reviewer 3 for addressing this point: we agree that this section needed a better introduction and we have tried to clarify this in the manuscript (Results section 3.2). While all TRIB proteins have been implicated in cancer, the role of TRIB2 in blood/circulating cells is more relevant than in adherent cells. Furthermore, the N- and C-Terminal domains of Tribbles are far less conserved between TRIB1/2 and 3 when compared to the well-conserved pseudo-kinase domain. Therefore understanding the specific roles of these 2 domains can lead to specific therapy approaches for individual Tribbles without the undesired effects of affecting any of the other Tribbles proteins.
“Make sure that gene/protein nomenclature is correct throughout the manuscript. E.g. TRIB3 (Human protein) and Trib3 (mouse protein) are mixed in the figures and text, but I assume all refer to human in this context. Or are mouse and human protein both used in this manuscript? Please clarify.”
The gene/protein nomenclature has been revised and mistakes have been corrected.
“In light of TP53 and CDKN1A binding, are there any effects on cell proliferation or cell death after TRIB1/2/3 expression in 293T or MCF7 cells?”
This is a valid point made by the reviewer and supported by very recent data from Orea-Soufi et al. (Cancers, 2021). It should be noted that TRIB1/2/3 effects on proliferation were observed upon long-term overexpression of knockdown, while we performed short-term (24h) doxycycline-mediated induction of TRIB expression. Within this timeframe, we did not observe differences in proliferation/cell death as assessed by MTS assays (data now shown). This is now indicated in the manuscript.
“In the manuscript text comparisons are made between the 293T and MCF7 data but there is no table or venn-diagram included to facilitate this comparison. Please include a table and venn that describes the overlap between TRIB1/3 binders in 293T vs. MCF7.”
We have added Venn diagrams showing the common interactors of TRIB1 and TRIB3 in HEK293T and MCF7 cells (Supplementary figure 2).

Round 2
Reviewer 1 Report
The revisions have improved presentation, and clarity. While I still have some doubts about the biological relevance of the reporter assay, the conclusions have been suitably explained.